# GABA and Proline Application Induce Drought Resistance in Oilseed Rape

**DOI:** 10.3390/plants14060860

**Published:** 2025-03-10

**Authors:** Sigita Jurkonienė, Virgilija Gavelienė, Rima Mockevičiūtė, Elžbieta Jankovska-Bortkevič, Vaidevutis Šveikauskas, Jurga Jankauskienė, Tautvydas Žalnierius, Liudmyla Kozeko

**Affiliations:** 1Laboratory of Plant Physiology, Nature Research Centre, Akademijos Str. 2, 08412 Vilnius, Lithuania; rima.mockeeviciute@gamtc.lt (R.M.); elzbieta.jankovska@gamtc.lt (E.J.-B.); vaidevutis.sveikauskas@gamtc.lt (V.Š.); jurga.jankauskiene@gamtc.lt (J.J.); tautvydas.zalnierius@gamtc.lt (T.Ž.); liudmyla.kozeko@gmail.com (L.K.); 2Department of Cell Biology and Anatomy, M.G. Kholodny Institute of Botany, National Academy of Sciences of Ukraine, Tereshchenkivska Str. 2, 01601 Kyiv, Ukraine

**Keywords:** *Brassica napus*, γ-aminobutyric acid, recovered growth, water deficit, drought-sensitive gene expression

## Abstract

This study investigates the effects of γ-aminobutyric acid (GABA) and proline, both individually and in combination, on the growth of oilseed rape under drought stress and following the resumption of irrigation. The goal was to determine whether the exogenous application of these compounds enhances the plants response to prolonged water deficit and, if so, to identify the biochemical processes involved in the plant tissue. The experiment was conducted under controlled laboratory conditions. After 21 days of plant cultivation, at the 3–4 leaf stage, seedlings were sprayed with aqueous solutions of GABA (0.1 mM) and proline (0.1 mM). The plants were then subjected to 8 days of severe drought stress, after which irrigation was resumed, and recovery was assessed over 4 days. The results showed that both amino acids alleviated the drought-induced stress as indicated by higher relative water content (RWC), increased levels of endogenous proline and photosynthetic pigments in leaves, and enhanced survival and growth recovery after drought. GABA-treated plants maintained membrane integrity and preserved plasma membrane (PM) ATPase activity during prolonged drought stress while reducing ethylene, H_2_O_2_, and MDA levels. Proline also influenced these biochemical responses, though to a lesser extent. The combination of GABA and proline facilitated better recovery of oilseed rape compared to the drought control group following rewatering. Notably, GABA treatment resulted in a significant increase in gene expression compared to the untreated control. Molecular analysis of drought-responsive genes revealed that the gene expression in plants treated with both proline and GABA was typically intermediate between those treated with proline alone and those treated with GABA alone. Based on these findings, we propose that GABA application could serve as an alternative to proline for improving oilseed rape’s drought tolerance, potentially increasing both crop yield and quality.

## 1. Introduction

Drought stress adversely affects plant growth and development, disturbing many physiological and biochemical processes. Oilseed rape is highly sensitive to drought and global climate change, which is causing severe and prolonged droughts in some parts of the world and is expected to reduce the survival and productivity of oilseed rape [1]. In Central and Eastern Europe, decreasing summer rainfall and increasing warm temperature extremely hamper plant growth and development [2]. Numerous studies have indicated that the exogenous application of hormones, polyamines, nutrients, antioxidants, osmoprotectants, etc., improves plant drought tolerance [3,4,5,6]. According to these studies, such compounds can enhance parameters such as morphology, photosynthetic capacity, and RWC, and promote osmolyte accumulation.

In recent years, the role of amino acids in the induction of plant stress resistance has received significant attention in both fundamental and applied fields of plant molecular physiology. More studies have shown that the exogenous application of the proteogenic amino acid proline was successful in enhancing plant resistance to various abiotic stresses. The use of proline as an osmotic protection against water deficit in wheat plants promotes the accumulation of high levels of chlorophyll, proline, glycine betaine, and soluble phenols [7,8,9]. Furthermore, we find evidence that γ-aminobutyric acid (GABA) is associated with plant stress response and the control of plant resistance to abiotic stresses [10,11,12]. In addition, GABA is a non-protein amino acid involved in a variety of physiological processes including protective effects on plants against drought stress, increasing osmolyte content and leaf turgor, and reducing oxidative damage [12,13]. Thus, proline, a proteogenic amino acid that acts as an osmoprotectant, subcellular stabilizer, and antioxidant, and GABA, a non-protein amino acid that acts as a signaling molecule, both play an important role in the protection of the cells in the event of water shortage. Another common feature of these amino acids is that their main biosynthetic pathway is from glutamate. In addition, the literature shows that treatment with exogenous GABA can improve the drought tolerance of cultivated plants (white clover, sunflower, cucumber) by increasing the free proline content, the activity of the enzymes peroxidase and catalase, the rate of photosynthesis, and the efficiency of water use [14,15]. Thus, GABA plays a positive role in physiological regulation when plants are exposed to unfavorable environmental stressors; hence, regulating GABA levels could be a promising approach to improve plant stress tolerance. At present, there is evidence that endogenous GABA concentrations in plants are relatively low but can increase in response to a range of stress conditions [13]. Thus, exogenously applied low doses of GABA could be an economically relevant and novel tool to induce drought tolerance. However, the effect of GABA application on drought tolerance in commercially important oilseed rape has not been investigated. Therefore, one of the questions of this study is as follows: Does the exogenous application of GABA improve the response of oilseed rape to prolonged water deficit and, if so, by which biochemical processes is this reflected in the plant tissues?

Investigations of crop plants show various adaptive and acclimatization strategies to drought stress, which range from seemingly simple morphological or physiological traits that serve as important stress tolerance markers to major upheavals in gene expression in which many transcription factors are induced [16,17]. Moreover, crop plant adaptation to drought is a very complex process, altering the expression level of numerous genes. The factors mitigating drought stress (GABA, proline) should change the expression of drought-induced genes under water deprivation conditions. So, such genes could be a valuable marker for the evaluation of drought stress diminishing. Recently a set of genes, which alter their expression level in response to simulated drought, was identified and confirmed by qRT-PCR in oilseed rape, therefore bringing them as putative candidates for water stress tolerance [18]. Among them, a strong increase in expression showed a *KIN2-like* gene which is similar to *LEA* (late embryogenesis abundant) and *KIN1/2* genes. It is known that genes encoding LEA proteins are expressed in seeds and vegetative organs during abiotic stress [19]. Also, the expression of the *KIN2* gene in *Arabidopsis* strongly increases in response to drought [20]. Two other genes, which expression increased in response to water deficit in oilseed rape, were genes encoding peroxygenase 3 (caleosin 3) and *EDL3* (EID1-like F-box protein 3) [18,21] acting via the proteasome degradation mechanism [22]. Also, the increase in expression in response to simulated drought in oilseed rape showed the gene of bZIP transcription factor *ABI5* [18]. The transcription factor is mostly active in the earliest stages of plant development and is important under abiotic stress conditions, mainly drought and salt [23]. *ABI5* could be used as a marker gene in drought tolerance studies if significant changes in expression could be detected under drought conditions compared to the untreated amino acid control. Taken together, the existing knowledge led us to the decision to evaluate *KIN2-like*, peroxygenase 3 (caleosin 3), *EDL3*, and *ABI5* gene expression after proline and GABA and their combined treatments in the adaptation of oilseed rape to drought, which have never been studied. Thus, to increase drought tolerance in oilseed rape, the effect of GABA and proline and their co-treatment on the induction of resistance was investigated.

## 2. Results

### 2.1. Selection of Active GABA Concentrations

The RWC of plants treated with 3 and 12.5 mL of 0.1 mM GABA per pot were 58% and 60%, respectively, while the RWC of drought-only treated plants on the 8th day of drought was 42% and 38% (high stress). It should be noted that all concentrations of GABA had no significant effect on the RWC of the leaves of continuously watered plants. Thus, the treatment with 0.1 mM GABA at 12.5 mL per pot resulted in less wilting than the drought control (Table 1). The choice of GABA concentration and dose to improve oilseed rape growth under simulated prolonged drought conditions showed that the most suitable concentration was 0.1 mM at 12.5 mL per pot (Table 2). It was observed that this dose of GABA increased the final average weight and survival of the recovered seedlings after 4 days of renewed irrigation (Table 1 and Table 2). Thus, treatment with 0.1 mM GABA at 12.5 mL per pot was chosen for further experiments as it promotes plant RWC and growth recovery of drought stressed plants.

### 2.2. Selection of Active Mix GABA + Proline Concentrations

Two variants of the mix were selected for complex treatment of proline 1 mM + GABA 0.1 mM and proline 0.1 mM and GABA 0.1 mM on fresh weight of oilseed rape seedlings (per plant) (Table 3) and on plant survival after 8 days of prolonged drought recovered after 12 days of irrigation (Table 3). Treatment with 0.1 mM GABA and 0.1 mM proline 12.5 mL per pot was selected for further experiments because it promoted drought-stressed plant growth recovery.

Various cytological, physiological, and biochemical indicators, such as RWC, ethylene emission, H_2_O_2_ level, MDA content, photosynthetic pigment levels, PM ATPase activity, and endogenous proline content, can be used to determine an externally unmanifested response to unfavorable environmental factors.

### 2.3. Impact of Two Amino Acids the RWC of Oilseed Rape Leaves Exposed to Prolonged Drought

The RWC of plant leaves exposed to drought for 4 to 8 days gradually decreased from 60% to 39%, indicating significant stress in the plants. Exogenous application of proline, GABA, and a combination of proline + GABA effectively mitigated drought-induced wilting of oilseed rape leaves (Figure 1). A positive effect of all applied compounds on the RWC of the plants was also observed after a 4-day recovery period. No significant differences in RWC were observed among the treatments when oilseed rape was grown under favorable conditions.

### 2.4. Impact of Two Amino Acids on Photosyntetic Pigment Content in Oilseed Rape Leaves Exposed to Prolonged Drought

The chlorophyll *a* and *b* content in rapeseed leaves significantly decreased under drought stress, being 26% and 47% lower after 4 days of drought and 57% and 67% lower after 8 days of drought compared to the corresponding control plants. The carotenoid content also decreased significantly when the plants were exposed to drought stress. The concentration of carotenoids dropped to 0.29 mg g^−1^ FW and 0.23 mg g^−1^ FW on drought days 4 and 8, respectively, when compared to 0.34 mg g^−1^ FW recorded in the leaves of irrigated plants at the same time. In plants exposed to exogenous GABA, proline and a mixture of these amino acids before drought stress, a higher content of chlorophyll *a*, chlorophyll *b*, and carotenoids were detected both during drought and recovery after drought (Table 4).

### 2.5. Ethylene

The ethylene emission study showed that during drought stress, the ethylene production in oilseed rape plants increased significantly, reaching 114.79 nl g^−1^ h^−1^ on the 8th day of drought, compared to 59.83 nl g^−1^ h^−1^ in the irrigated control. Proline and GABA compounds applied alone significantly reduced ethylene formation during drought stress: 82.70 and 60.53 nl g^−1^ h^−1^ on the 8th day of drought, respectively. After irrigation resumed, ethylene emission of the proline-treated plants, and especially in the GABA-treated plants, decreased much faster and reached the levels of the irrigated control plants within 4 days (Figure 1).

### 2.6. Impact of Two Amino Acids on Biochemical Responses of Oilseed Rape Exposed to Prolonged Drought

#### 2.6.1. H_2_O_2_

H_2_O_2_ concentrations increased significantly on days 4 and 8 of the drought, reaching 8.66 and 16.03 nmol g^−1^ FW, respectively, compared to 4.07–4.79 nmol g^−1^ FW in the well-watered control plants (Figure 1). Pre-drought treatment of oilseed rape plants with GABA, proline, and a mixture of proline + GABA resulted in a reduction in H_2_O_2_ content. On the eighth day of the drought, the lowest H_2_O_2_ content (10.48 nmol g^−1^ FW) was recorded in oilseed rape plants treated with GABA before the drought. All the treatments had a positive effect, and the oilseed rape plants recovered from the drought stress.

#### 2.6.2. MDA

The accumulation of MDA in rapeseed leaves increased continuously during drought, from 20.96 nmol g^−1^ FW detectable on the 4th day of drought to 38.28 nmol g^−1^ FW on the 8th day. Meanwhile, lipid peroxidation in the drought-stressed oilseed rape plants sprayed with GABA prior to stress was the lowest, with MDA levels of 16.41 and 24.39 nmol g^−1^ FW on days 4 and 8 (Figure 1). In the proline + GABA treatment, lower MDA content was observed only on the 8th day of drought compared to plants affected only by drought. As plants recovered from drought, a similar decrease in MDA levels was recorded in plants treated with amino acids and their mixture.

#### 2.6.3. Free Proline

The accumulation of free endogenous proline in plants exposed to 4 days of drought stress was 8.32 µmol g^−1^ FW and after 8 days of drought, it increased to 12.38 µmol g^−1^ FW. In contrast, the control plants had proline levels of 0.61 µmol g^−1^ FW on day 4 and 0.68 µmol g^−1^ FW on day 8 (Figure 1). After 8 days of drought, the highest amounts of free proline were detected in the variants sprayed with GABA and proline with levels of 16.6 and16.3 µmol g^−1^ FW, respectively. The results showed that free proline content significantly increases as the drought period continues and decreases with the resumption of irrigation.

#### 2.6.4. PM ATPase Activity

Drought stress slightly increased PM ATPase activity on the fourth day following the initiation of drought treatment. Notably, plants sprayed with GABA exhibited similar enzyme activity to those not exposed to drought. However, on the 8th day of the drought, the PM ATPase activity in oilseed rape leaves was significantly reduced, except in the GABA-treated plants, which showed a 53% increase compared to the drought control reaching 20.01 μmol P_i_ mg^−1^ protein h^−1^. On the fourth day after irrigation resumed, the PM ATPase activity in the GABA- and proline-treated plants increased by 44% and 56%, respectively, compared to the drought control, and was significantly higher than that of the continuously irrigated plants (Figure 1).

#### 2.6.5. Effect of Amino Acids on Expression Levels of Four Genes in Oilseed Rape Exposed to Prolonged Drought

After 4 and 8 days of drought, the expression of all four genes was increased in the drought-affected oilseed rape plants without additional treatments compared to the watered control. The weakest increase in expression was observed in the *ABI5* gene after eight days of drought treatment, while the *KIN2-like* gene showed a notably strong upregulation after 4 days of drought stress (Figure 2). Amino acids such as proline, GABA, and a combination of these reduced the effect of drought on the expression of studied genes. Additionally, differences in gene expression levels were observed at different points after treatment. Proline showed minimal increase in the expression of all four genes tested, compared to the corresponding watered control (except *EDL3*, where expression increased by 3.7 times) after four days of drought. Meanwhile, the GABA-treated plants at the same time point showed increased gene expression compared to the corresponding watered control and the expression was consistently significantly higher than in the drought-affected proline-treated plants. After eight days of drought treatment, a strong increase in gene expression was observed in the proline-treated plants compared to the corresponding watered and untreated drought controls and it always exceeded gene expression levels in the GABA-treated plants. This occurred even though the increase in gene expression in the GABA-treated plants was almost always higher after eight days, compared to the four days of drought stress (except for caleosin/peroxygenase 3 gene expression).

#### 2.6.6. Impact of Exogenous GABA and Proline on Plant Survival

After 8 days of simulated drought stress and 4 days of irrigation recovery, the number of oilseed rape plants treated with GABA and proline was significantly higher (on average 42%) compared to the plants treated with drought stress alone. The number of plants treated with both GABA and proline was lower than that of plants treated with either GABA or proline alone and these differences were statistically significant (Figure 3).

## 3. Discussion

Prolonged drought reduces the relative water content of plants and leads to various metabolic and photosynthetic disturbances [13,24]. In response to drought stress, the accumulation of some amino acids, such as GABA and proline, can provide plant protection against drought damage by maintaining cellular water balance and cell membrane stability through osmotic regulation. In addition, as drought limits cell growth due to a reduction in turgor pressure, this abiotic stress can affect the formation of above-ground plant mass [15]. In the current study, we showed that water deficit negatively affected the fresh weight (FW) parameters of oilseed rape shoots and was dependent on prolonged drought. Shoot FW was significantly higher when drought-stressed plants were sprayed with 0.1 mM GABA or 1 mM proline. According to the literature, the RWC is considered an indicator of plant water status and dehydration [25,26]. It predicts the physiological state of the plant and indicates the plant’s ability to retain water under adverse conditions [27,28], so, it can be used as a criterion for evaluating the effects of various compounds used to enhance plant drought tolerance [29]. In our study, RWC was a primally reliable indicator for evaluating the effect of GABA and proline in response to prolonged drought of oilseed rape. The results showed that the RWC of drought-stressed oilseed rape leaves was significantly reduced, which is in agreement with other studies [30,31,32,33]. Under well-watered conditions, foliar treatment with amino acids did not affect leaf RWC, but it significantly increased by 30% and 58% in the GABA- and proline + GABA-treated plants after 4 and 8 days of prolonged drought, respectively. At 4 days of rewatering, the RWC of previously drought-stressed plants increased to the nonstress level, which was not significantly different among the GABA, proline, and GABA + proline treatments, indicating that all plants were fully rehydrated on rewatering.

Numerous studies have shown that oxidative stress caused by water deficit significantly reduces photosynthesis and pigments in plants [29,34,35,36]. This was confirmed by our results, where chlorophyll *a* and *b* content in oilseed rape leaves decreased to 0.54 mg g^−1^ FW and 0.11 mg g^−1^ FW after 8 days of drought, while in the irrigated control, it was 1.27 mg g^−1^ and 0.33 mg g^−1^ FW, respectively. Moreover, the exogenous application of GABA and proline before the drought resulted in a significant improvement of photosynthetic pigments, not only during prolonged drought but also during the recovery of the plants from the resumption of watering. This confirms the results of other researchers, where exogenous GABA treatment significantly improved chlorophyll and carotenoid content in pepper [29], while proline treatment improved chlorophyll content in the leaves of wheat [9], barley [37], and rice [38] under drought conditions.

Furthermore, several researchers found that the decrease in chlorophyll content was due to the overproduction of reactive oxygen species (ROS), degradation in nutritional balance and inactivation of enzyme activities, as well as a decline in cellular water contents [30,39]. H_2_O_2_ is ROS produced by cellular metabolism and is an indicator of the ROS scavenging capacity of plants under stress. It is often found in the literature that plants under drought stress produce ROS such as H_2_O_2_ [40,41]. Our study showed that H_2_O_2_ concentration increased 3.3-fold during 8 days of drought compared to the watered control. Similar drought-induced oxidative stress was reported in other authors’ studies on oilseed rape plants [31,33]. In the current study, pre-drought spraying of oilseed rape with GABA, proline, and proline + GABA resulted in a decrease in H_2_O_2_ content under prolonged drought simulation and had a positive effect on plant recovery resuming irrigation. Meanwhile, we find evidence that exogenous stress protecting compounds significantly reduced H_2_O_2_ accumulation under drought stress conditions [33,42]. Excessive concentrations of free radicals, including H_2_O_2_, can cause damage to cell membranes, ion leakage, and osmotic imbalances, so it is essential to maintain their levels [30,31]. The results indicate that GABA might have a role in preventing oxidative damage caused by the increased accumulation of H_2_O_2_ during prolonged drought stress.

In addition, we recorded high levels of H_2_O_2_ in drought-stressed oilseed rape plants, leading to an increase in the production of MDA, an indicator of oxidative stress. The accumulation of MDA in oilseed rape leaves increased steadily during the drought, from 20.96 nmol g^−1^ FW detected on the 4th day of the drought to 38.28 nmol g^−1^ FW on the 8th day of the drought. These findings were consistent with the results obtained in the literature, which indicate that drought induces the overproduction of ROS and oxidative stress in a wide range of plant species. According to Saha et al. [43], a significant increase in MDA content was observed in rice plants after the 8th day of drought. Hasanuzzaman et al. [31] monitored the extent of lipid peroxidation (MDA content) in drought-stressed oilseed rape plants. There are several studies that exogenous amino acids (GABA, proline) can effectively reduce MDA levels during drought stress in cultivated plants [10,44]. Our studies showed that the exogenous treatment of oilseed rape with GABA or proline resulted in lower oxidative damage: on the 4th day of drought, MDA levels were 22% and 16% lower than in the control plants, respectively. As the drought persisted longer, lipid peroxidation was significantly lower in the leaves of both the proline (31%)- and GABA (36%)-treated plants, alone and in their combination (21%).

Several reports suggest that PM ATPase plays a key role in plant growth and development and is an important component of membrane integrity in response to abiotic stress [45,46,47]. There is also contrary evidence in the literature indicating that H^+^-ATPase activity can increase [48] or decrease [47] when plants are exposed to drought stress. Furthermore, we find relevant data that the treatment of leaves with stress-protective compounds such as polyamines significantly increased PM ATPase activity by 43.79% compared to the untreated plants under drought stress. The results of our study showed that H^+^-ATPase activity was significantly reduced in the drought control samples. Also, the 8th day of drought resulted in a significant decrease in the PM ATPase activity of rapeseed leaves, except GABA treated plants, which increased its activity by an average of 53% compared to the drought control, reaching 20.01 μmol P_i_ mg^−1^ protein h^−1^. On the fourth day of irrigation renewal, the PM ATPase activity of GABA- and proline-treated plants increased by 44% and 56%, respectively, compared to the drought control and was significantly higher than that of the continuously irrigated plants. Furthermore, it was found that the complex application of canola with GABA and proline had a positive effect on the H^+^-ATPase activity, but it was still lower.

A range of osmotically active molecules such as proline accumulated to balance the water relations under drought stress and, in this case, is most extensively studied osmolyte [49,50,51]. Numerous studies have reported that the exogenous application of proline has shown positive effects on plants growing under drought stress, promoting growth and antioxidant activity [52,53]. Moreover, we find data that the exogenous application of amino acid GABA can increase free endogenous proline levels in plant tissues [54], but the effects of this amino acid on oilseed rape in improving resistance to drought have not been investigated. In our research, we found that exogenous GABA treatment reduced the concentration of proline in rapeseed leaves under drought stress, while after a prolonged drought, plants sprayed with exogenous GABA had a significantly higher endogenous proline content than the untreated plants. It was also found that the plants exposed to exogenous proline, according to the amount of free proline in the leaf tissues, were slightly lower than those exposed to GABA. The results on the free proline content showed that the proline content significantly increases as the drought period continues and decreases with the start of watering. Therefore, we proposed the positive effects of exogenous GABA and proline on alleviating the oxidative stress associated with proline accumulation and homeostasis in oilseed rape plants under drought conditions.

Several publications indicate that under drought stress conditions, increased levels of ethylene occur in many plant species [55,56]. Ethylene production after drought stress can reduce photosynthesis, inhibit root growth, and reduce shoot/leaf expansion [57]. Our data contribute to the suggestion that exogenous amino acids affecting ethylene content may help to remove the inhibitory effect of drought stress on plant growth. Ethylene emission analysis showed that during drought stress, the amount of ethylene produced by rapeseed plants increased intensively and reached 114.79 nl g^−1^ h^−1^ on the 8th day of drought, while in the irrigated control plants, it was 59.83 nl g^−1^ h^−1^. Meanwhile, GABA or proline significantly reduced ethylene formation during drought stress. After resuming watering, the ethylene emission of the GABA-treated plants decreased much faster and reached the level of irrigated plants within 4 days. A mixture of proline + GABA increased ethylene accumulation but was still lower than in the drought control. These data agreed with the results obtained by Sharma et al. [58] that changes in ethylene content during drought can activate the plant antioxidant defense system, resulting in a reduction in oxidative stress, with accompanied recovery of plant growth and photosynthetic efficiency.

Drought stress affects and alters the expression of many genes. The four putative drought-responsive genes, *KIN2-like*, caleosin/peroxygenase 3, EDL 3 (EID 1 like F-box protein 3), and *ABI 5*, were chosen as a marker to understand the patterns of action of proline and GABA on the genetic level in the response of rapeseed to drought stress. It is known that these genes increase their expression in response to drought stress conditions [18,20,21,22,59]. As expected, the expression of all four genes was increased in the simulated drought-affected oilseed rape plants with no additional treatments compared to the watered control. After 4 days of drought, remarkably strong upregulation occurred in the *KIN2-like* gene, which is consistent with the data obtained by Zhang et al. [19]. After eight days of drought, the increase in *ABI5* gene expression was weak. This is consistent with our obtained data showing that plants without additional treatment suffered from drought after 4 and 8 days of drought application. Proline, GABA, and their mix reduced the effect of drought. Interestingly, although both amino acids diminished the effect of drought to similar levels, the response at the gene expression level differed between the proline- and GABA-treated plants. It seems that proline and GABA act differently under drought conditions at the genetic level. It is known that proline can protect plants from environmental stresses in multiple ways: acting as an osmolyte and molecular chaperone, through ROS scavenging activity, or as a donor of electrons in mitochondria [60]. So, it can be speculated that the application of exogenous proline to rapeseed initially enhances the proline metabolic pool inside, which in turn increases plant drought tolerance to such a level, that plants do not need to increase the expression of the genes necessary for resistance to drought. But later, exogenous proline’s drought-protective properties weaken, and the drought-dependent genes are upregulated. The induction of drought-responsive genes could be promoted also by proline catabolism, which could be enhanced in the case of exogenous proline application. Indeed, it was shown that some products of proline catabolism could induce osmotically regulated genes in rice [61]. The different situation is with GABA. Although GABA, like proline, can protect plants from drought by acting through various physiological and biochemical mechanisms [12], it seems that such GABA action is not enough for plants to be protected against water deficit, so the genes responsible for drought resistance were upregulated already in the early stages of drought. On the other hand, after eight days of drought, the lower expression level of drought-induced genes of GABA treated plants could indicate that GABA functioning as a drought-protective molecule at physiological and biochemical levels may continue longer compared to proline. So, in this case, the plants did not need to maintain very strong drought-inducible gene expression to achieve the same protection against drought. Finally, in the plants treated with proline and GABA simultaneously, the gene expression level was almost always intermediate between the gene expression in the proline-treated and GABA-treated plants separately. It seems that their action is not additive. In the recovery phase, the expression of the tested genes was often at low levels, not even reaching the level of the corresponding watered control, indicating that the tested genes are not important for plant recovery. The exception was the *KIN2-like* gene, which expression was surprisingly high in the untreated plants recovering from drought. Interestingly, the GABA-treated plants’ gene expressions in the recovery phase were almost always slightly higher compared to the other variants, though this may not be related in the recovery from drought conditions.

Thus, in oilseed rape, various molecular, biochemical, physiological, and morphological processes are damaged under drought stress, but the plant response to amino acid application can induce biochemical and cellular changes in the tissues that ensure survival. We found that exogenous GABA and proline improved the survival of oilseed rape seedlings after simulated drought stress, and the number of plants surviving after resumed irrigation was significantly higher compared with plants exposed to drought alone.

## 4. Materials and Methods

### 4.1. Research Object and Growth Conditions

Winter oilseed rape (*Brassica napus* L.) cv. “Visby” seeds were sown in pots with a peat moss (pH 5.5–6.5). The plants were grown under controlled conditions in a Climacell plant growing chamber (Medcenter Einrichtungen, Planegg, Germany) at 23 °C and a 16/8 h (light/darkness) photoperiod at a light intensity of 75 μmol m^−2^ s^−1^. Soil moisture was maintained at ~70%. The moisture was measured with a soil moisture meter (Biogrod, Shanghai, China).

### 4.2. Treatments

#### 4.2.1. Spraying with Amino Acids

Aqueous solutions of amino acids L-proline (Carl ROTH, Karlsruhe, Germany), GABA (Sigma-Aldrich, Darmstadt, Germany), and a mixture of L-proline and GABA (Sigma-Aldrich) were used to spray the seedlings at the 3–4 leaf stage, BBCH scale 13–19 [62]. The concentration range was chosen based on literature data and our previous experiments [63,64,65]. Spraying of the plants was carried out using a hand-held sprayer, at a dose of 3 mL and 12.5 mL in each pot. The control plants were sprayed with the same volume of distilled water.

#### 4.2.2. Drought Simulation

For the drought stress control studies, the plants were subjected to prolonged drought stress for 8 days to reach a high water deficit. During the simulated drought, irrigation was interrupted to allow the soil to dry out gradually. The soil moisture was 40% on the fourth drought day, 20% on the eight drought day, and 70% on the fourth plant recovery day.

#### 4.2.3. Irrigation Renewal

After 8 days of prolonged drought, irrigation was resumed to achieve a soil moisture content of 70%. Plant recovery was assessed after 4 days.

### 4.3. Determination of Active GABA Concentration

In order to determine the concentration of active GABA, oilseed rape seeds were sown in (10 × 10 × 12) cm pots with peat substrate. Each experimental unit consisted of 14 seeds. The pots without GABA were the control. Three treatments were used to study the effect of GABA: (1) control, H_2_O 3 mL per pot; (2) GABA 0.1 mM 3 mL per pot; (3) GABA 1 mM 3 mL per pot; (4) GABA 10 mM·3 mL per pot; (5) control, H_2_O 12.5 mL per pot; (6) GABA 0.1 mM 12.5 mL per pot; (7) GABA 1 mM 12.5 mL per pot; (8) GABA 10 mM·12.5 mL per pot; the other eight treatments were the same plus the drought treatment. The seedlings were foliar sprayed with GABA solutions at the BBCH 13–19 in the third to fourth leaf stage, and the control seedlings were sprayed with water.

### 4.4. Experimental Scheme

Eight pots (40 × 17 × 13) cm were used in the experiment, four of which were used for irrigation and four for drought imitation. The experiment was carried out three times, with sixty plants in each pot. Aqueous solutions of L-proline (0.1 mM), GABA (0.1 mM), and both amino acids in the same concentration (0.1 mM) were used, 12.5 mL per pot. The spray regime was as follows: (1) control with irrigation, (2) proline with irrigation, (3) GABA with irrigation, (4) proline + GABA with irrigation, (5) drought, (6) proline with drought, (7) GABA with drought, and (8) proline + GABA with drought (Figure 4).

### 4.5. Sampling

Plant leaves were sampled three times for testing: on day 4 of drought (at 40% soil moisture), on day 8 of drought (at 20% soil moisture), and on day 4 of plant recovery after watering (at 70% soil moisture). At the same time, the watered plants (control) were sampled. Thirty seedlings were selected for morphometric measurements. For biochemical analysis, three independent replicates were carried out using fully expanded leaves of the oilseed rape plants. Freshly taken leaf samples were analyzed for ethylene emission and pigment content. For H^+^-ATPase activity, MDA, H_2_O_2_, and proline assays, the samples were flash frozen in liquid nitrogen and stored at −80 °C in a low-temperature freezer (Skadi Green line, EU) until further analysis. The number of surviving seedlings was counted immediately after 41 days of cultivation.

### 4.6. Determination of RWC

The RWC was determined according to Hodges et al. [66] with modifications as presented in [28]. Leaves from the same part of each plant were picked and weighed (FW), then placed in distilled water for 24 h to allow the leaves to become saturated with water and weighed again (SW), then wrapped in craft paper, dried at 80 °C to a constant weight, and the dry weight was determined (DW). The RWC of the leaves was determined according to the following formula:RWC (%) = (FW − DW)/(SW − DW) × 100%,(1)

### 4.7. Determination of Plant Survival

After resumed watering, plant survival, i.e., the proportion of plants that retained the ability to grow at the stem apex, was determined.

### 4.8. Assessment of Biochemical Parameters

#### 4.8.1. Photosynthetic Pigments

The photosynthetic pigments were extracted from fresh leaves using *N*,*N*′-dimethyl-formamide (DMF) (Sigma-Aldrich). Light absorption was measured at 480, 664, and 647 nm. Chlorophyll *a* and *b* contents were calculated according to the method described by Wellburn [67].

#### 4.8.2. MDA Content

The MDA content was determined using the method described by Li [68]. A 0.3 g leaf sample was ground in a diluent with 5 mL of 1% trichloroacetic acid (TCA). The extract was then centrifuged at 10,000× *g* for 5 min at 4 °C. Then, 0.3 mL of the supernatant was mixed with 1.2 mL of the MDA solution, which contained 20% TCA and 0.5% thiobarbituric acid (TBA). The mixture was placed in a water bath at 95 °C for 30 min, immediately cooled, and centrifuged again at 10,000× *g* for 10 min. The absorbance of the supernatant was measured with a spectrophotometer at 532 nm.

#### 4.8.3. H_2_O_2_ Content

The H_2_O_2_ content of the leaves was measured using the method of Alexieva et al. [69]. Then, 0.15 g of the leaves was ground in a diluent with 5 mL of 0.1% (*v*/*v*) TCA. The extract was centrifuged at 10,000× *g* for 5 min. Then, 25 μL of the supernatant was mixed with 250 μL of 100 mM phosphate buffer (pH 7) and 500 μL of 1 M potassium iodide (KI). Absorbance was measured with a spectrophotometer at 390 nm.

#### 4.8.4. Ethylene Emission

Ethylene emission from freshly harvested leaves was estimated using the method of Child et al. [70]. Weighed leaf samples were placed in 40 mL clear glass vials (Agilent technology, Santa Clara, CA, USA) sealed with PTFE/Si septa caps and incubated for 24 h at 21 °C in the dark. After incubation, 1 mL of the gas sample from each vial was withdrawn with a gas-tight syringe (Agilent technology) and injected into a gas chromatograph equipped with a stainless steel column (PROPAC R, Waltham, MA, USA) and a hydrogen flame ionization detector. The injector, column, and detector temperatures were 110, 90, and 150 °C, respectively. Helium (AGA, Vilnius, Lithuania) was used as the carrier gas. Calibration was carried out with ethylene standard (Messer, Bridgewater, NJ, USA). The results are expressed in nl g^−1^ FW h^−1^.

#### 4.8.5. H^+^-ATPase Activity Assay

A membrane-enriched fraction of microsomes was isolated from the plant samples. The protein content was determined by the Bradford dye coupling method with absorbance measured at 595 nm. The H^+^-ATPase activity of the microsomal fraction was assessed by the isolation of inorganic phosphate (P_i_), which accumulates as a result of ATP hydrolysis [71]. The P_i_ color reaction was carried out using ammonium molybdate and stannous chloride and the absorbance measured at 750 nm. The enzyme activity was expressed as μmol P_i_ mg^−1^ protein h^−1^.

#### 4.8.6. Proline

Free proline content was assessed with a ninhydrin-based method using a cuvette spectrophotometer (Analytik Jena Specord 210 Plus, Jena, Germany). Proline was extracted by heating plant aliquots of ground plant material (0.5 g) for 20 min in pure ethanol. The resulting mixture was left overnight at 4 °C and then centrifuged at 14,000× *g* (5 min). The color reaction mixture consists of ninhydrin 1% (*w*/*v*) in acetic acid 60% (*v*/*v*) and ethanol 20% (*v*/*v*) [72]. The absorbance was read spectrophotometrically at 520 nm. The corresponding content of proline was determined using the standard curve. The results were expressed as µmol of proline µmol g^−1^ FW.

### 4.9. Molecular Techniques

#### 4.9.1. RNA Extraction and Reverse Transcription

Total RNA was extracted from 150 mg of plant leaf material using a commercial RNA isolation kit Quick-RNA Plant MiniPrep™ Kit (ZymoResearch, Irvine, CA, USA) following the manufacturer’s recommendations. To avoid contamination with genomic DNA, the extracted total RNA was treated with DNase I using the RapidOut DNA Removal Kit (Thermo Scientific, Waltham, MA, USA). The concentration and purity of the treated RNA was evaluated with the spectrophotometer NanoPhotometer P330 (IMPLEN, Westlake Village, CA, USA). The DNase-treated RNA samples were reverse-transcribed using the Verso cDNA Synthesis Kit (Thermo Scientific, Waltham, MA, USA) following the manufacturer’s recommendations.

#### 4.9.2. Real-Time Quantitative PCR

Real-time quantitative PCR was carried out using the Maxima SYBR Green qPCR Master Mix (2×) kit (Thermo Scientific) following the manufacturer’s recommendations. The cycling conditions comprised one cycle at 95 °C for 2 min and 40 cycles at 95 °C for 15 s followed by 60 °C for 1 min. PCR amplification was performed in an Azure Cielo (Bio-Rad, Dublin, CA, USA) real-time PCR machine. Gene expression was calculated using the 2^−ΔΔCt^ method [73].

#### 4.9.3. Primers

The sequences of the primers used in the work were taken from [18]. Two, under different experimental conditions, commonly used reference genes of actin 7 (ACT7) and ubiquitin-conjugating enzyme 21 (UBC21) [74], were used for gene expression analysis. The sequences of all the primers used in the work are listed in Table 5.

### 4.10. Statistical Analysis

The results are expressed as the mean ± standard deviation (SD) of three independent experiments with at least three replicates. The data were analyzed using analysis of variance (ANOVA). Tukey’s test was used to assess the statistical significance of the differences between the means (*p* < 0.05). The post hoc Duncan’s Multiple Range Test (DMRT) was used to compare the means of the different groups after ANOVA.

## 5. Conclusions

Exogenously added amino acids (GABA and proline) reduced drought-induced damage to oilseed rape. GABA and proline applied alone or in combination alleviated drought stress in oilseed rape by improving water status, photosynthetic pigments and osmoprotective substance accumulation, and membrane integrity in leaf tissue. Plants exposed to amino acids produced large amounts of endogenous proline when exposed alone. In this study, different effects of GABA, proline, and GABA + proline on plant drought tolerance and recovery from drought were found. GABA contributed mainly to the enhancement of plant tolerance to drought stress, whereas proline was more effective in promoting drought recovery compared to GABA. The tested amino acids reduced the simulation of ethylene emission in drought-stressed seedling leaves. A higher amount of ethylene emission was observed under GABA + proline exposure. Molecular studies of drought-responsive genes revealed that GABA and proline can protect plants from drought by acting through *KIN2-like*, caleosin/peroxygenase 3, *EDL 3*, and *ABI 5* gene expression. Therefore, the application of GABA is an alternative to the proline technique for improving oilseed rape tolerance to drought and could be used to increase the amount of crops.

## Figures and Tables

**Figure 1 plants-14-00860-f001:**
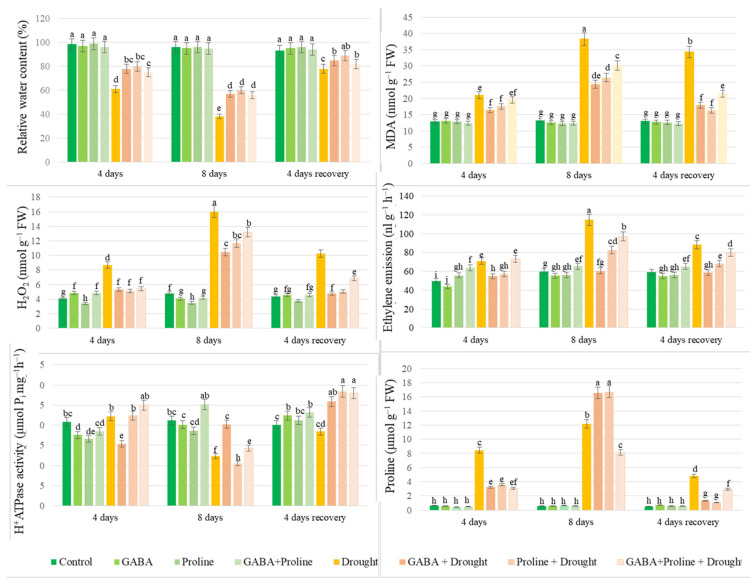
Impact of GABA and proline application on RWC, ethylene emission, H_2_O_2_ level, MDA content, PM ATPase activity, and endogenous proline in oilseed rape leaves following simulated prolonged drought and recovery. Vertical error bars represent standard deviation of mean of three replications (*n* = 3). Different lowercase letters indicate statistically significant differences (*p* < 0.05). The order of the data is given in the legend.

**Figure 2 plants-14-00860-f002:**
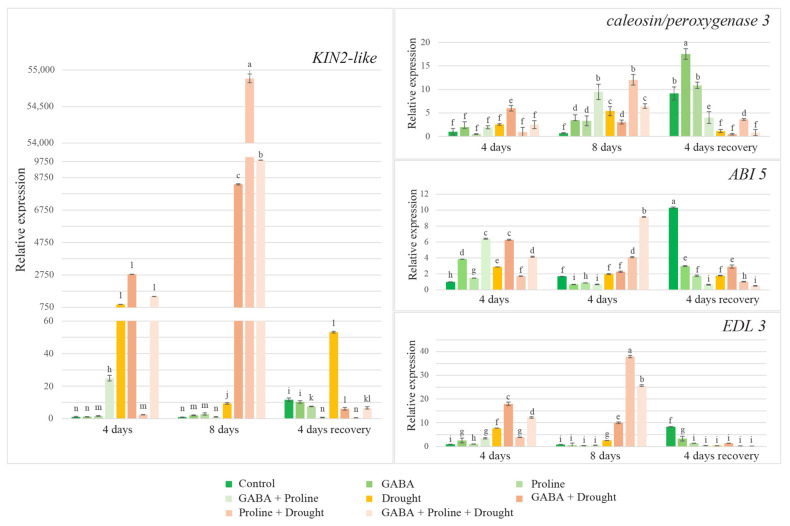
The effect of exogenous GABA and proline application on the expression of drought-sensitive gene oilseed rape under prolonged drought conditions. Different lowercase letters indicate statistically significant differences (*p* < 0.05).

**Figure 3 plants-14-00860-f003:**
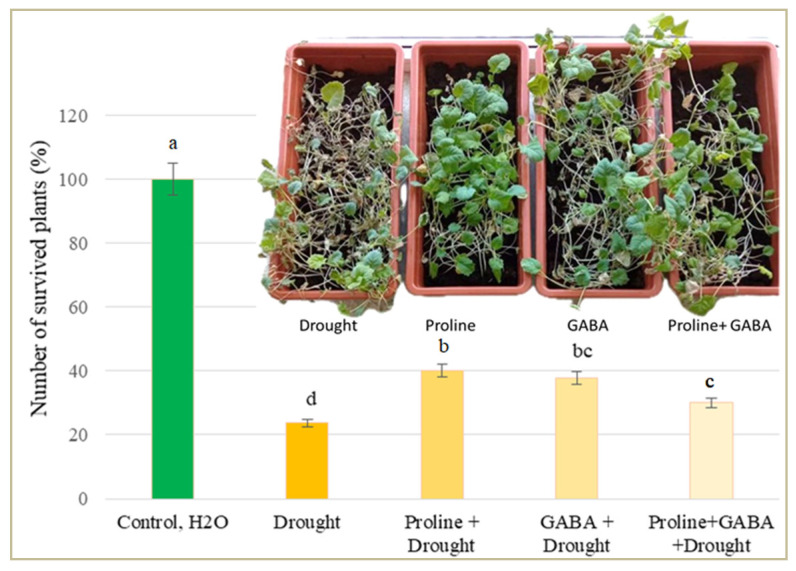
Effect of exogenous GABA and proline on *Brassica napus* survival after 8 days of prolonged drought recovery following irrigation. Different lowercase letters indicate statistically significant differences (*p* < 0.05).

**Figure 4 plants-14-00860-f004:**
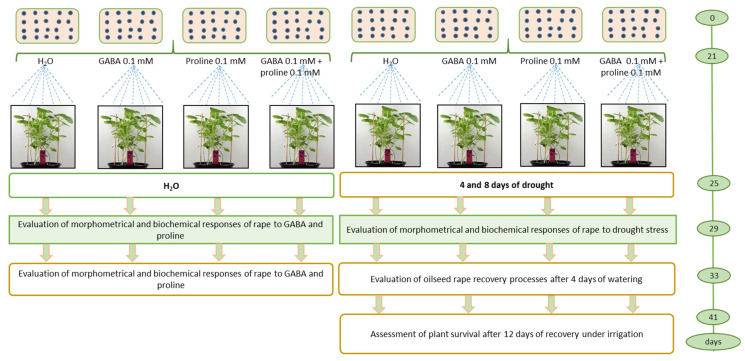
Experiment design. Sixty plants per pot, with three replicates per variant and three biological experiments.

**Table 1 plants-14-00860-t001:** Effect of GABA treatment on RWC of leaves with continuous increase in water deficit (simulating drought) and on fresh weight of oilseed rape recovering from 8 days prolonged drought after 4 days of irrigation.

Treatment (3 mL)	RWC, %		Average Fresh Weight Per Plant	
	4 Days	8 Days	g	%
Control, H_2_O	76.8 ± 0.1 ab	79.1 ± 5.3 a	0.97 ± 0.31 a	100 a
GABA 0.1 mM	78.1 ± 2.6 a	77.6 ± 2.1 ab	0.97 ± 0.30 a	100 a
GABA 1 mM	76.9 ± 5.0 ab	80.7 ± 2.5 a	0.83 ± 0.38 c	86 c
GABA 10 mM	74.4 ± 2.8 b	73.0 ± 0.8 b	0.95 ± 0.50 ab	98 ab
Drought	62.9 ± 2.4 e	42.8 ± 1.0 d	0.68 ± 0.27 c	100 c
GABA 0.1 mM + Drought	71.1 ± 0.9 c	58.4 ± 1.5 c	0.82 ± 0.18 a	121 a
GABA 1 mM + Drought	67.2 ± 3.0 d	56.1 ± 8.2 c	0.79 ± 0.17 b	116 b
GABA 10 mM + Drought	65.9 ± 1.6 de	54.3 ± 7.0 c	0.83 ± 0.17 a	122 a
Treatment (12.5 mL)				
Control, H_2_O	76.2 ± 3.2 a	77.0 ± 1.3 ab	1.13 ± 0.34 ab	100 ab
GABA 0.1 mM	76.6 ± 4.0 a	80.2 ± 2.0 a	1.18 ± 0.28 a	104 a
GABA 1 mM	75.4 ± 4.0 a	79.8 ± 2.0 a	1.18 ± 0.32 a	104 a
GABA 10 mM	75.4 ± 1.6 a	80.1 ± 3.5 a	1.16 ± 0.44 a	103 a
Drought	60.4 ± 6.9 bc	38.0 ± 6.2 e	0.78 ± 0.52 c	100 c
GABA 0.1 mM + Drought	63.5 ± 5.1 b	60.0 ± 5.1 c	0.99 ± 0.41 a	127 a
GABA 1 mM + Drought	59.3 ± 2.1 c	56.5 ± 2.1 cd	0.91 ± 0.41 b	117 b
GABA 10 mM + Drought	63.1 ± 2.5 b	54.0 ± 2.5 d	0.93 ± 0.42 b	119 b

Mean (±SE) was calculated from three replicates for each treatment. Values in column with different letters are significantly different at *p* ≤ 0.05 applying Duncan’s Multiple Range Test (DMRT). Average fresh weight control (H_2_O) and drought were referred to in test results and were marked as 100%.

**Table 2 plants-14-00860-t002:** Effect of GABA on *Brassica napus* plant survival after 8 days of prolonged drought recovered after 12 days of irrigation.

Treatment (12.5 mL)	Number of Survived Plants (%)
Control, H_2_O	100 a
GABA 0.1 mM	100 a
GABA 1 mM	100 a
GABA 10 mM	100 a
Drought	25 d
GABA 0.1 mM + Drought	83 bc
GABA 1 mM + Drought	75 c
GABA 10 mM + Drought	67 c

Mean was calculated from three replicates for each treatment. Values in column with different letters are significantly different at *p* ≤ 0.05 applying DMRT.

**Table 3 plants-14-00860-t003:** Effect of complex treatment of proline 1 mM + GABA 0.1 mM and proline 0.1 mM + GABA 0.1 mM on fresh weight of oilseed rape seedlings (per plant) under 8 days simulated drought conditions and survival after 8 days of prolonged drought recovered after 12 days of irrigation.

Treatment (12.5 mL)	Average Fresh Weight Per Plant		Number of Survived Plants (%)
	g	%	
Control, H_2_O	1.64 ± 0.57 b	100 b	100 a
Proline 1 mM + GABA 0.1 mM	1.68 ± 0.66 b	102 b	100 a
Proline 0.1 mM + GABA 0.1 mM	1.90 ± 0.69 a	116 a	100 a
Drought	0.69 ± 0.31 c	100 c	27 d
Proline 1 mM + GABA 0.1 mM + Drought	0.93 ± 0.29 b	135 a	64 c
Proline 0.1 mM + GABA 0.1 mM +Drought	1.43 ± 0.50 a	121 b	85 b

Mean (±SE) was calculated from three replicates for each treatment. Values in column with different letters are significantly different at *p* ≤ 0.05 applying DMRT. Average fresh weight control (H_2_O) and drought were referred to in test results and were marked as 100%.

**Table 4 plants-14-00860-t004:** Effect of GABA and proline treatment on photosynthetic pigment content in oilseed rape seedlings under drought stress.

Treatment	Pigment Contents (mg g^−1^ FW)								
	Chlorophyll *a*			Chlorophyll *b*			Carotenoids		
	4 Days	8 Days	4 Days Recovery	4 Days	8 Days	4 Days Recovery	4 Days	8 Days	4 Days Recovery
Control, H_2_O	1.24 a	1.27 a	1.28 a	0.38 a	0.33 a	0.34 a	0.34 a	0.34 b	0.34 a
GABA	1.27 a	1.25 a	1.23 a	0.37 a	0.34 a	0.32 a	0.36 a	0.37 a	0.35 a
Proline	1.28 a	1.26 a	1.22 a	0.37 a	0.32 a	0.35 a	0.35 a	0.37 a	0.34 a
GABA + Proline	1.28 a	1.24 a	1.29 a	0.34 a	0.32 a	0.32 a	0.36 a	0.36 a	0.35 a
Drought	0.92 c	0.54 c	0.75 d	0.20 c	0.11 c	0.16 c	0.29 b	0.23 c	0.25 b
GABA + Drought	1.16 b	0.72 b	1.09 b	0.27 b	0.17 b	0.27 b	0.35 a	0.31 b	0.34 a
Proline + Drought	1.18 b	0.70 b	1.04 b	0.29 b	0.16 b	0.26 b	0.38 a	0.32 b	0.35 a
GABA + Proline + Drought	1.14 b	0.75 b	0.99 bc	0.27 b	0.17 b	0.24 b	0.32 a	0.33 b	0.34 a

Mean was calculated from three replicates for each treatment. Values in column with different letters are significantly different at *p* ≤ 0.05 applying DMRT.

**Table 5 plants-14-00860-t005:** The sequences of the primers used in the work.

Gene Locus in *Brassica napus*	Gene Name	Forward Primer Sequence (5′ → 3′)	Reverse Primer Sequence (5′ → 3′)
BnaA03g27910D	*KIN2-like*	GCCAGACTAAGGAGAAGACAAGT	TGTTCTTGTTCATGCCGGTTT
BnaA05g10200D	*caleosin*/*peroxygenase 3*	TGCCATCACCATTACTGCCT	CGGTTCCCCTCGGTTAAGTT
BnaC04g09800D	*EDL 3* (*EID 1 like F-box protein 3*)	GTGGGTGGTGGCGGAGAA	CCGCCTGTCTCCTCACGA
BnaA05g08020D	*ABI 5*	TAAGCAGCCGAGTCTTCCAC	ACCACCGCCGTTATTAGCAT
BnaA02g00190D	*ACT7*	CCTCTCAACCCGAAAGCCAA	CATCACCAGAGTCGAGCACA
BnaA06g27860D	*UBC21*	TATCCTCTGCAGCCTCCTCA	CTGTCTGCCTCAGGATGAGC

## Data Availability

The data supporting the reported results can be found in the archive of scientific reports of the Nature Research Centre.

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
