# Peer review of "GABA and Proline Application Induce Drought Resistance in Oilseed Rape"

_plants, 2025, doi:10.3390/plants14060860_

Round 1

Reviewer 1 Report

Comments and Suggestions for Authors

The manuscript “GABA and Proline Application Induce Drought Resistance of Oilseed Rape” describe the study of amino acids - GABA and Proline- implies in drought stress control in a specie with economic impact.  GABA is a ubiquitous amino acid whose physiological role has been widely studied and known, as proline too. However, the use of molecular markers makes original this work.

English style and grammar should be reviewed in the abstract and result sections. Many issues and sentence without coherence must be reviewed. Particularly, many sentences are too long and turn difficult the comprehension in the Abstract.

Summarizing, the manuscript could be divided in 2 parts, the first one where the effects of drought stress on various physiological markers of rape is described and the second one, the molecular study.

The first one is repetitive and unoriginal, many papers and reviews described the results obtained in this manuscript. However, this research is valuable and contributes to the field, particularly the four putative drought responsive genes chosen as markers. Hence, the manuscript could be shortened bearing in mind that many results presented are obvious and/or previously described and others are originals.

Author Response

Response to Reviewer 1 Comments

Point 1: The manuscript “GABA and Proline Application Induce Drought Resistance of Oilseed
Rape” describe the study of amino acids - GABA and Proline- implies in drought stress control
in a specie with economic impact. GABA is a ubiquitous amino acid whose physiological role
has been widely studied and known, as proline too. However, the use of molecular markers
makes original this work.

Response 1: Thank You for your comment. It is a great pleasure to get such a comment.

Point 2: English style and grammar should be reviewed in the abstract and result sections. Many
issues and sentence without coherence must be reviewed. Particularly, many sentences are too
long and turn difficult the comprehension in the Abstract.

Response 2: Thank You for your comments. We have revised the abstract and all text according
to your recommendation. The manuscript was edited by native speaker editor too.

Point 3: Summarizing, the manuscript could be divided in 2 parts, the first one where the effects
of drought stress on various physiological markers of rape is described and the second one, the
molecular study.

Response 3: Thank you for your advice. We edited the text according to your recommendations.
We have made many of corrections to shorten the text.

Point 4: The first one is repetitive and unoriginal, many papers and reviews described the results
obtained in this manuscript. However, this research is valuable and contributes to the field,
particularly the four putative drought responsive genes chosen as markers. Hence, the manuscript
could be shortened bearing in mind that many results presented are obvious and/or previously
described and others are originals.

Response 4: Thank you for your comment. We edited the text according to your
recommendations. In the revised manuscript, we have focused on the novelty of this study, as
changes in the expression of drought-responsive genes in oilseed rape after treatment with
GABA, proline and GABA+proline have not been identified so far.

Reviewer 2 Report

Comments and Suggestions for Authors

Comments: In this study, exogenous GABA and proline improved the survival of oilseed rape seedlings after simulated drought stress, and the number of plants surviving after resumed irrigation was significantly higher compared with plants exposed to drought alone, suggesting that application of GABA is an alternative to proline technique for improving oilseed rape tolerance to drought. However, this work can be improved by including and discussing some of the following topics:

1.     The subject of title is large. Please revise.

2.     Please simplify the description of the abstract.

3.     The research progress of GABA and Proline in the introduction section need to update.

4.     In results, significance is needed. For example, table 1, table 3, etc.

5.     In results, why did you only choose to measure to RWC, ethylene emission, H2O2 level, MDA content, PM ATPase activity, and endogenous proline content? What's the connection among them.

6.     In results, the figures are blurry.

7.     Line 555, How does the “amount and quality of crops” reflect?

Author Response

Response to Reviewer 2 Comments

Point 1: In this study, exogenous GABA and proline improved the survival of oilseed rape
seedlings after simulated drought stress, and the number of plants surviving after resumed
irrigation was significantly higher compared with plants exposed to drought alone, suggesting
that application of GABA is an alternative to proline technique for improving oilseed rape
tolerance to drought. However, this work can be improved by including and discussing some of
the following topics: 1) The subject of title is large. Please revise.

Response 1: Thank you for your valuable comment. We slightly edited the title of the paper to
“GABA and Proline Application Induce Drought Resistance in Oilseed Rape”.

Point 2: Please simplify the description of the abstract.

Response 2: Thank you for your comment. We carefully edited the text of the abstract according
to your recommendations.

Point 3: The research progress of GABA and Proline in the introduction section needs to update.

Response 3: Thank you for your valuable comment. We revised the research progress of GABA
and Proline in the introduction section according to your recommendations.

Point 4: In results, significance is needed. For example, table 1, table 3, etc.

Response 4: Thank you for your valuable comment. We specified the values in a column with
different letters are significantly different. Also, we explained that the average fresh weight of
control (H2O) and Drought were marked as 100%l and were compared to other weight results.

Point 5: In results, why did you choose to measure to RWC, ethylene emission, H2O2 level,
MDA content, PM ATPase activity, and endogenous proline content? What's the connection
among them.

Response 5: Thank you for the comment. We have added text to the manuscript explaining our
choice to measure physiological drought indices and describing the connection among them.

Point 6: In results, the figures are blurry.

Response 6: Thank you for your valuable comment. We sharpened the figure colours and text of
the axes.

Point 7: Line 555, How does the “amount and quality of crops” reflect?

Response 7: Thank You for the comment. We have edited the text on crop yields and quality.

Reviewer 3 Report

Comments and Suggestions for Authors

This manuscript investigated the influence of γ -aminobutyric acid (GABA) and proline alone and in combination on the growth of oilseed rape under drought stress, confirming that the exogenous application of these two compounds can improve the tolerance of oilseed rape to drought stress. However, many previous studies have reported that the treatment of exogenous GABA and proline can help various cultivated plants to improve the drought tolerance such as wheat, maize, barley, white clover, sunflower, and cucumber. Therefore, the results achieved from the present study are fewer novels. Additionally, the whole manuscript is not well written in body structure and language, and there are many mistakes in grammar, syntax, word spelling, and format throughout the text, tables, and figures. Authors need to comprehensively revise this submission for publication.

Comments on the Quality of English Language

The whole manuscript is not well written in body structure and language, and there are many mistakes in grammar, syntax, word spelling, and format throughout the text, tables, and figures. Authors need to comprehensively revise this submission for publication.

Author Response

Response to Reviewer 3 Comments

Point 1: This manuscript investigated the influence of γ -aminobutyric acid (GABA) and proline
alone and in combination on the growth of oilseed rape under drought stress, confirming that the
exogenous application of these two compounds can improve the tolerance of oilseed rape to
drought stress. However, many previous studies have reported that the treatment of exogenous
GABA and proline can help various cultivated plants to improve the drought tolerance such as
wheat, maize, barley, white clover, sunflower, and cucumber. Therefore, the results achieved
from the present study are fewer novels. Additionally, the whole manuscript is not well written
in body structure and language, and there are many mistakes in grammar, syntax, word spelling,
and format throughout the text, tables, and figures. Authors need to comprehensively revise this
submission for publication.

Response 1: Thank You for your valuable comments. We have edited the full text of the article,
the titles of the tables and figures, and have made a complete overhaul to bring it up to date and
relevant.

Point 2: The whole manuscript is not well written in body structure and language, and there are
many mistakes in grammar, syntax, word spelling, and format throughout the text, tables, and
figures. Authors need to comprehensively revise this submission for publication.

Response 2: Thank you for your comment. We corrected mistakes and edited the text according
to your recommendations. 

Reviewer 4 Report

Comments and Suggestions for Authors

Dear authors, after reading your paper, I have some comments:

Abstract 

"The study aims  aimed  (suggest) to investigate whether γ-aminobutyric acid (GABA) and proline, alone and in combination, can influence the growth of oilseed rape under drought stress."

Introduction

Overall, the introduction effectively builds a strong scientific rationale for the study. It aligns with academic standards by presenting a straightforward research question, relevant background, and a well-defined knowledge gap. Minor improvements in economic relevance and novelty discussion could further enhance its impact.

Material and methods

-The units must be unified: mmol or mM

- Figure 4: GABA + Proline 0.1 mMM

- Text proline was used in the figures Proline 

Results 

The results demonstrate that GABA and proline improve drought tolerance in oilseed rape through multiple mechanisms, including higher water retention, reduced oxidative damage, enhanced gene expression, and improved survival rates. GABA is particularly effective in early drought response by maintaining membrane integrity, ROS balance, and rapid gene activation. Proline plays a delayed but substantial role in long-term stress adaptation. The combined treatment does not provide additive effects, suggesting that GABA and proline share overlapping pathways. These findings provide valuable agricultural insights and support the potential use of GABA as a drought-protective treatment in crops. 

Discussion and conclusions are more than appropriate and suggest that proline’s effects are delayed because it first accumulates in plant tissues before activating stress-related genes. In contrast, GABA triggers an immediate stress response, possibly by regulating stress signalling pathways.

Overall, this paper should be published with minor revisions.

Author Response

Response to Reviewer 4 Comments

Point 1: Abstract. "The study aims aimed (suggest) to investigate whether γ-aminobutyric acid
(GABA) and proline, alone and in combination, can influence the growth of oilseed rape under
drought stress..

Response 1: Thank you for your comment. We carefully edited the text of the abstract according
to your recommendations.

Point 2: Introduction. Overall, the introduction effectively builds a strong scientific rationale for
the study. It aligns with academic standards by presenting a straightforward research question,
relevant background, and a well-defined knowledge gap. Minor improvements in economic
relevance and novelty discussion could further enhance its impact.

Response 2: Thank you for your comment. It is a great pleasure to get such comment. We
stressed on economic relevance and novelty in introduction according to your recommendations.

Point 3: Material and methods. The units must be unified: mmol or mM.

Response 3: Thank you for your valuable comment. We edited the text to unify concentration
units.

Point 4: Figure 4: GABA + Proline 0.1 mMM

Response 4: Thank you for the comment. We have corrected this mistake.

Point 5: Text proline was used in the figures Proline

Response 5: Thank you, we corrected this mistake.

Point 6: The results demonstrate that GABA and proline improve drought tolerance in oilseed
rape through multiple mechanisms, including higher water retention, reduced oxidative damage,
enhanced gene expression, and improved survival rates. GABA is particularly effective in early
drought response by maintaining membrane integrity, ROS balance, and rapid gene
activation. Proline plays a delayed but substantial role in long-term stress adaptation. The
combined treatment does not provide additive effects, suggesting that GABA and proline share
overlapping pathways. These findings provide valuable agricultural insights and support the
potential use of GABA as a drought-protective treatment in crops.

Response 6: Thank you for your positive comments.

Point 7: Discussion and conclusions are more than appropriate and suggest that proline’s effects
are delayed because it first accumulates in plant tissues before activating stress-related genes. In
contrast, GABA triggers an immediate stress response, possibly by regulating stress signalling
pathways.

Response 7: Thank you for such a nice comment and new ideas for the future.

Reviewer 5 Report

Comments and Suggestions for Authors

The review of the paper entitled “GABA and Proline Application Induce Drought Resistance of Oilseed Rape” submitted to Plants, MDPI, by Jurkoniene et al. The paper focus on biochemical and molecular response of Brassica napus plants submitted to drought stress and the modulation of this response by exogenous applied proline and GABA. The advantage of this work is to used combined treatment of proline and GABA to deal with stress effect in plants. The paper is very nicely prepared and written in proper English. All the results are clearly presented and discussed with the literature. The literature cited is up-to-date, which is extremely important. The conclusions are cautious but well-supported by the evidence presented in the manuscript. I have just a few minor remarks.

L46, please add full name for RWC

Gene names should be italics (introduction, 2.6.5. ), also Latin name of species such as Arabidopsis should be italics.

Table 1 – please add information that control with water and drought was also used as a control which you referred to the rest results, that is why they are mark as 100%. You should more precisely stated that treatment with water were compared to the “control” and drought treatment were compared to “drought”. The term control is confusing. Maybe you can mark as bold the control lines (this with 100%).

Section 2.2 You wrote that you measure recovery after 12 days of irrigation, whereas in section 4.5 you wrote just about 4 days of recovering. I see this information in section 4.4. However, in section 4.4 you mention 5 days of recovering and in sampling 4 days. Please modify it to make all of these sections coherent.

Figure 1. If it is still possible to modify the color code for this figure, please make it more color-blindness friendly. If not, please add the note, that the order of data are as follows in the legend.

In the term chlorophyll, the letters a and b should be italics.

L414 Please verify if the BBCH code is right, or should it be 13-14?

Drought simulation – did you monitored soil moisture during this 8-days period of drought? What was the change in soil moisture? Could you provide this information in this section?

The abbreviation list does not cover the gene names. Also, some of the full names are begun from upper case, but should from lower (RWC, TCA).

Author Response

Response to Reviewer 5 Comments

Point 1: The review of the paper entitled “GABA and Proline Application Induce Drought
Resistance of Oilseed Rape” submitted to Plants, MDPI, by Jurkoniene et al. The paper focus on
biochemical and molecular response of Brassica napus plants submitted to drought stress and the
modulation of this response by exogenous applied proline and GABA. The advantage of this
work is to used combined treatment of proline and GABA to deal with stress effect in plants. The
paper is very nicely prepared and written in proper English. All the results are clearly presented
and discussed with the literature. The literature cited is up-to-date, which is extremely important.
The conclusions are cautious but well-supported by the evidence presented in the manuscript. I
have just a few minor remarks.

Response 1: Thank You, for such a nice comment. It’s a great pleasure to get such comment.

Point 2: L46, please add full name for RWC

Response 2: Thank you for the comment. We added full name for RWC.

Point 3: Gene names should be italics (introduction, 2.6.5. ), also Latin name of species such
as Arabidopsis should be italics.

Response 3: Thank you for your valuable comment. Mistakes are corrected.

Point 4: Table 1 – please add information that control with water and drought was also used as a
control which you referred to the rest results, that is why they are mark as 100%. You should
more precisely stated that treatment with water were compared to the “control” and drought
treatment were compared to “drought”. The term control is confusing. Maybe you can mark as
bold the control lines (this with 100%).

Response 4: Thank you for your valuable comment. We edited the text under the table 1 and 3
indicating that treatment with water were compared to the “control” and drought treatment were
compared to “drought” and marked as bold the control lines according to your recommendations.

Point 5: Section 2.2 You wrote that you measure recovery after 12 days of irrigation, whereas in
section 4.5 you wrote just about 4 days of recovering. I see this information in section 4.4.
However, in section 4.4 you mention 5 days of recovering and in sampling 4 days. Please modify
it to make all of these sections coherent..

Response 5: Thank you for your valuable comment. We edited the figure of experimental design
and changed 5 days of recovery to 4 days. The survival of oilseed rape was measured after 12
days of recovery by irrigation.

Point 6: Figure 1. If it is still possible to modify the color code for this figure, please make it
more color-blindness friendly. If not, please add the note, that the order of data are as follows in
the legend.

Response 6: Thank you for your valuable comment. We modified the colours for figure 1 the
pointed that the order of data are as follows in the legend according to your recommendations.

Point 7: In the term chlorophyll, the letters a and b should be italics.

Response 7: Thank you for your comment. We edited the style of the font.

Point 8 L414 Please verify if the BBCH code is right, or should it be 13-14?

Response 8: Thank you for your valuable comment. We corrected BBCH to 13–19.

Point 9: Drought simulation – did you monitored soil moisture during this 8-days period of
drought? What was the change in soil moisture? Could you provide this information in this
section?

Response 9: Thank you for your valuable comment. We provided the information on soil
moisture during 8-days drought period in drought simulation subsection according to your
recommendations.

Point 10: The abbreviation list does not cover the gene names. Also, some of the full names are
begun from upper case, but should from lower (RWC, TCA).

Response 10: Thank you for your valuable comment. We edited the abbreviation list according
to your recommendations.

Round 2

Reviewer 2 Report

Comments and Suggestions for Authors

Accept.

Author Response

Dear Dr. Soo-In Sohn and Dr. Subramani Pandian

With this resubmission, please find our revised manuscript (ID: plants-3443189) for consideration to be published in the Special Issue “Advances in Molecular Genetics and Breeding of Brassica napus L.” of Plants.

We are grateful for the valuable and constructive comments of the reviewers. We have carefully revised the manuscript. Hopefully, the subsequent modifications are satisfying for all the reviewers. Please find our responses to all the comments and recommendations of reviewers below. We look forward to Your positive response.

Sincerely,

Drs Virgilija Gavelienė and Sigita Jurkonienė

Nature Research Centre, Institute of Botany

Akademijos Str. 2, Vilnius LT-08412, Lithuania

phone: +370 5 2729047

E-mail: virgilija.gaveliene@gmail.com; sigita.jurkoniene@gamtc.lt

Response to Reviewer 3 Comments

Point 1: Some subheadings in Results section such as "2.3 RWC" and "2.4 Photosyntetic pigments" are too simple, and authors need to rewrite them for clear meaning.

Response 1: Thank You for Your valuable comments. For clarity, we have adjusted the subheadings of the Results section according to your recommendations.

Point 2: In Figure 4, all the pictures with dots or tobacco plants are completely the same, which is very monotonous. I would suggest authors to modify this figure

Response 2: Thank You for Your comment. We have corrected the caption of Figure 4 to "Experimental design". It then became clear why all the seeds (dots) are exactly the same at the start of the experiment. We apologise for the mistake.

Point 3: English and description need to be improved further

Response 3: Thank you for your comment. We carefully edited the text according to your recommendations.

Reviewer 3 Report

Comments and Suggestions for Authors
  1. Some subheadings in Results section such as "2.3 RWC" and "2.4 Photosyntetic pigments" are too simple, and authors need to rewrite them for clear meaning.
  2. In Figure 4, all the pictures with dots or tobacco plants are completely the same, which is very monotonous. I would suggest authors to modify this figure.  
  3. English and description need to be improved further.
Comments on the Quality of English Language

English and description need to be improved further.

Author Response

(The authors gave the same response as above.)
